# Sample-efficient diffusion-based control of complex nonlinear systems

## Abstract

Complex nonlinear system control faces challenges in achieving sample-efficient, reliable performance. While diffusion-based methods have demonstrated advantages over classical and reinforcement learning approaches in long-term control performance, they are limited by sample efficiency. This paper presents SEDC (Sample-Efficient Diffusion-based Control), a novel diffusion-based control framework addressing three core challenges: high-dimensional state-action spaces, nonlinear system dynamics, and the gap between non-optimal training data and near-optimal control solutions. Through three innovations - Decoupled State Diffusion, Dual-Mode Decomposition, and Guided Self-finetuning - SEDC achieves 39.5%-49.4% better control accuracy than baselines while using only 10% of the training samples, as validated across three complex nonlinear dynamic systems. Our approach represents a significant advancement in sample-efficient control of complex nonlinear systems. The implementation of the code can be found at https://anonymous.4open.science/r/DIFOCON-C019.

## 1. Introduction

The control of complex systems plays a critical role across diverse domains, from industrial automation (Baggio et al., 2021) and biological networks (Gu et al., 2015) to robotics (Zhang et al., 2022). Given the challenges in deriving governing equations for empirical systems, data-driven control methods—which design control modules directly based on experimental data collected from the system, bypassing the need for explicit mathematical modeling—have gained prominence for their robust real-world applicability (Baggio et al., 2021; Janner et al., 2022; Ajay et al., 2022; Zhou et al., 2024; Liang et al., 2023; Wei et al., 2024).

LBefore data-driven control methods, traditional Proportional-Integral-Derivative (PID) (Li et al., 2006) controllers dominated complex system control through continuous error correction. However, these classical methods show limitations with complex nonlinear systems due to their linear control nature. Data-driven machine learning approaches have emerged to address these limitations by learning nonlinear control policies from interaction data, falling into three categories: supervised learning, reinforcement learning (RL), and diffusion-based methods. Supervised learning approaches like Behavior Cloning (BC) (Pomerleau, 1988) learn direct state-to-action mappings from expert demonstrations, while RL methods like Batch Proximal Policy Optimization (BPPO) (Zhuang et al., 2023) learn control policies through value function approximation, showing better adaptability to high-dimensional states than classical methods. However, both approaches often exhibit myopic decision-making in long-horizon tasks due to their iterative view of control dynamics. In contrast, diffusion-based methods (Janner et al., 2022; Ajay et al., 2022; Zhou et al., 2024; Liang et al., 2023; Wei et al., 2024) reformulate control as sequence generation, enabling comprehensive optimization over entire system trajectories. This long-term perspective allows diffusion-based methods to overcome limitations of both classical and RL approaches, achieving superior long-term control performance.

The success of diffusion models in data-driven control stems from their exceptional ability to learn complex trajectory distributions from empirical data. In practice, these trajectories are typically collected from systems operated under empirical rules or random policies. Moreover, due to operational costs, the available data volume is often limited. Diffusion-based methods must therefore learn effective control policies from such non-optimal and sparse trajectory data—a challenge that manifests in three key aspects. **First, limited data volume impedes sample-efficient learning in high-dimensional systems.** Existing diffusion-based controllers (e.g., DiffPhyCon (Wei et al., 2024)) attempt to directly generate long-term ($T$ steps) state-action trajectories by learning a $T \times (P + M)$-dimensional distribution of system states $y^P$ and control inputs $u^M$. This joint distribution implicitly encodes system dynamics of state transitions under external control inputs, which often leads to physically inconsistent trajectories when training samples are insufficient. **Second, learning control policies for nonlinear systems remains an open challenge both theoretically and practically.** Traditional analytical methods (Baggio et al., 2021) designed for linear systems

fail to perform robustly when applied to nonlinear systems. While diffusion-based approaches (Janner et al., 2022; Ajay et al., 2022; Zhou et al., 2024) employ deep neural networks (e.g., U-Net architectures) as denoising modules to capture nonlinearity, learning effective control policies from limited data remains particularly challenging for complex systems with strong nonlinearity, such as fluid dynamics and power grids. **Third, extracting improved control policies from non-optimal training data poses fundamental difficulties.** Diffusion-based methods (Janner et al., 2022) struggle when training data significantly deviates from optimal solutions. Although recent work (Wei et al., 2024) introduces reweighting mechanism to expand the solution space during generation, discovering truly near-optimal control policies remains elusive without explicit optimization guidance.

To address these challenges, we propose SEDC (**S**ample-**E**fficient **D**iffusion-based **C**ontrol), a novel diffusion-based framework for learning control policies of complex nonlinear systems with limited, non-optimal data. At its core, SEDC reformulates the control problem as a denoising diffusion process that samples control sequences optimized for reaching desired states while minimizing energy consumption. To address the curse of dimensionality, we introduce Decoupled State Diffusion (DSD), which confines diffusion process within state space and leverages inverse dynamics to generate control inputs, i.e., actions. This approach reduces learning complexity in high-dimensional systems while ensuring physics-aware control synthesis. To tackle strong nonlinearity, we propose Dual-Mode Decomposition (DMD) by designing a dual-UNet denoising module with residual connections. This architecture decomposes system dynamics into hierarchical linear and nonlinear components, enabling structured modeling of complex systems. To bridge the gap between non-optimal offline training data and optimal control policies, we introduce Guided Self-finetuning (GSF). This method progressively synthesizes guided control trajectories for iterative fine-tuning, facilitating exploration beyond initial training data and convergence toward near-optimal control strategies.

We demonstrate SEDC's superiority over traditional, reinforcement learning, and diffusion-based methods through experiments on three typical complex nonlinear systems. Our model demonstrates 39.5%-49.4% improvement in control accuracy compared to state-of-the-art baselines while maintaining better balance between accuracy and energy consumption. In sample efficiency experiments, SEDC matches state-of-the-art performance using only 10% of the training samples. Additional ablation studies validate the effectiveness of SEDC's key design components.

## 2. Related Works

**Classic control methods.** Data-driven control of complex systems has witnessed significant methodological developments across multiple paradigms. Classical control methods, represented by Proportional-Integral-Derivative (PID) controllers (Li et al., 2006), operate through continuous sensing-actuation cycles in a feedback-based manner. While these methods offer straightforward implementation, they face fundamental limitations when dealing with high-dimensional complex scenarios. More sophisticated analytical approaches, such as those presented in (Baggio et al., 2021), have attempted to determine optimal control inputs for complex networks without explicit dynamics knowledge. However, their foundation in linear systems theory inherently restricts their applicability to nonlinear systems.

**Data-driven control methods.** The emergence of supervised learning (Pomerleau, 1988) and reinforcement learning (Haarnoja et al., 2018; Zhuang et al., 2023) has introduced more adaptive approaches to complex control problems, demonstrating promising results in sequential decision-making tasks. However, these approaches often struggle with real-world deployment due to computational constraints and the challenge of making effective decisions over extended time horizons. More recently, denoising diffusion probabilistic models (Ho et al., 2020) have emerged as a powerful framework for modeling high-dimensional distributions, achieving remarkable success across various domains including image, audio, and video generation (Dhariwal & Nichol, 2021; Kong et al., 2020; Ho et al., 2022). This success has inspired their application to control problems, with several works demonstrating their potential in robotic control (Janner et al., 2022; Ajay et al., 2022) and trajectory generation (Liang et al., 2023; Zhou et al., 2024). The diffusion framework has also shown promise in related technical domains such as optimization (Krishnamoorthy et al., 2023; Sun & Yang, 2023) and inverse problems (Chung et al., 2022). For diffusion-based control methods, works like (Janner et al., 2022; Ajay et al., 2022) demonstrate capabilities in generating long-term control trajectories for reinforcement learning environments, but they employ generic architectures that struggle to capture highly nonlinear dynamics, while our method incorporates specialized designs for effective nonlinear system learning. Other works like (Liang et al., 2023; Zhou et al., 2024) focus primarily on robotic control without specific considerations for complex system dynamics, while our approach is specifically designed to handle the challenges of high-dimensional state-action spaces and strong nonlinearity in complex systems. Recent work DiffPhyCon (Wei et al., 2024) incorporates reweighting techniques to optimize trajectories beyond the training data distribution, and attempts to optimize trajectories through implicit dynamics modeling and reweighting mechanisms, but this approach lacks explicit guidance for

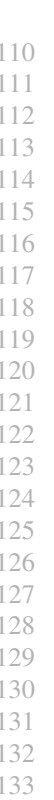
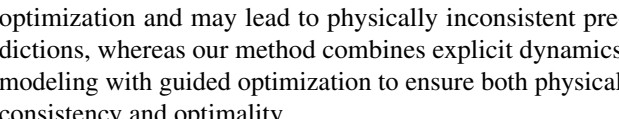

*Figure 1.* Illustration of SEDC, the proposed conditional diffusion-based controller.

optimization and may lead to physically inconsistent predictions, whereas our method combines explicit dynamics modeling with guided optimization to ensure both physical consistency and optimality.

## 3. Backgrounds

### 3.1. Problem Setting

The dynamics of a controlled complex system can be represented by the differential equation $\dot{\mathbf{y}}_t = \Phi(\mathbf{y}_t, \mathbf{u}_t)$, where $\mathbf{y}_t \in \mathbb{R}^N$ represents the observed system state and $\mathbf{u}_t \in \mathbb{R}^M$ denotes the control input. We assume the system satisfies the controllability condition without loss of generality: for any initial state $\mathbf{y}_0^*$ and target state $\mathbf{y}_f$, there exists a finite time $T$ and a corresponding control input $\mathbf{u}$ that can drive the system from $\mathbf{y}_0^*$ to $\mathbf{y}_f$. This assumption ensures the technical feasibility of our control objectives. In practical applications, beyond achieving state transitions, we need to optimize the energy consumption during the control process. The energy cost can be quantified using the L2-norm integral of the control input: $J(\mathbf{y}, \mathbf{u}) = \int_0^T |\mathbf{u}(t')|^2 dt'$. For data-driven optimal control problems, we can only understand the system dynamics through observational data. Consider a dataset $D = \{\mathbf{u}^{(i)}, \mathbf{y}^{(i)}\}_{i=1}^P$ containing $P$ non-optimal control trajectories, where each trajectory consists of:(1) complete state trajectories $\mathbf{y}^{(i)}$ sampled at fixed time intervals; (2) corresponding control input sequences $\mathbf{u}^{(i)}$. Based on this dataset, our objective is to find the optimal control

input trajectory $\mathbf{u}^* \in \mathbb{R}^{T \times M}$ that satisfies:

$$\begin{aligned}
\mathbf{u}^* &= \arg\min_{\mathbf{u}} J(\mathbf{y}, \mathbf{u}) \\
\text{s.t.} \quad & \Psi(\mathbf{u}, \mathbf{y}) = 0, \quad \mathbf{y}_0 = \mathbf{y}_0^*, \quad \mathbf{y}_T = \mathbf{y}_f,
\end{aligned} \quad (1)$$

where $\mathbf{y} \in \mathbb{R}^{T \times N}$ is the corresponding complete state trajectory given $\mathbf{y}_0$ and $\Psi(\mathbf{u}, \mathbf{y}) = 0$. Here, $\Psi(\mathbf{u}, \mathbf{y}) = 0$ represents the system dynamics constraint implicitly defined by dataset $D$. This constraint effectively serves as a data-driven representation of the unknown dynamics equation $\dot{\mathbf{y}}_t = \Phi(\mathbf{y}_t, \mathbf{u}_t)$.

Our key idea is to train a diffusion-based model to directly produce near-optimal control trajectories $\mathbf{u}_{[0:T-1]}$, providing a starting state $\mathbf{y}_0^*$, the target $\mathbf{y}_f$ and optimized by the cost $J$. Next, we summarize the details of the diffusion-based framework.

### 3.2. Diffusion Model

Diffusion models have become leading generative models, showing exceptional results across image synthesis, audio generation and other applications (Ho et al., 2020; Dhariwal & Nichol, 2021; Song & Ermon, 2019). These models, when applied to trajectory generation, operate by progressively adding noise to sequential data in the forward process and then learning to reverse this noise corruption through a denoising process. We denote that $\mathbf{x}^k$ represents the sequential data at diffusion timestep $k$. In the forward process, a clean trajectory $\mathbf{x}^0$ is progressively corrupted through $K$

timesteps, resulting in a sequence of increasingly noisy versions $\mathbf{x}^1, \mathbf{x}^2, ..., \mathbf{x}^K$. Each step applies a small amount of Gaussian noise:

$$q(\mathbf{x}^k|\mathbf{x}^{k-1}) = \mathcal{N}(\mathbf{x}^k; \sqrt{1-\beta^k}\mathbf{x}^{k-1}, \beta^k\mathbf{I}),$$

where $\beta^k$ is a variance schedule that controls the noise level. With a large enough $K$ we can get $q(\mathbf{x}^K) \approx \mathcal{N}(\mathbf{x}^K; \mathbf{0}, \mathbf{I})$. In the reverse process, the diffusion model learns to gradually denoise the data, starting from pure noise $\mathbf{x}^K$ and working backward to reconstruct the original plausible trajectory $\mathbf{x}^0$. Each denoising step is conditioned on the start and target state:

$$p_\theta(\mathbf{x}^{k-1}|\mathbf{x}^k, \mathbf{y}_0^*, \mathbf{y}_f) = \mathcal{N}(\mathbf{x}^{k-1}; \mu_\theta(\mathbf{x}^k, k, \mathbf{y}_0^*, \mathbf{y}_f), \mathbf{\Sigma}^k),$$

where $\theta$ represents the learnable parameters of the model and $\Sigma^k$ is from a fixed schedule.

**Training of diffusion model.** In order to facilitate the design of the denoising network, the network with $\theta$ for the denoising process does not directly predict $\mu$. Instead, it is trained to learn to predict clean trajectory $\mathbf{x}^0$, outputting $\hat{\mathbf{x}}^0$.

The training objective for diffusion models typically involves minimizing the variational lower bound (VLB) on the negative log-likelihood (Sohl-Dickstein et al., 2015). In practice, this often reduces to a form of denoising score matching (Song & Ermon, 2019):

$$\mathbb{E}_{\mathbf{x},k,\mathbf{y}_0^*,\mathbf{y}_f,\epsilon}[||\mathbf{x} - \mathbf{x}_\theta(\mathbf{x}^k, k, \mathbf{y}_0^*, \mathbf{y}_f)||^2],$$

where $\mathbf{x}, k, \mathbf{y}_0^*, \mathbf{y}_f$ are sampled from the dataset, $k \sim \mathcal{U}\{1, 2, ..., K\}$ is the step index and $\epsilon \sim \mathcal{N}(\mathbf{0}, \mathbf{I})$ is the noise used to corrupt $\mathbf{x}$.

## 4. SEDC: the Proposed Method

In this section, we introduce our three key innovative designs of SEDC: Decoupled State Diffusion, Denoising Network Design of Dual Mode Decomposition, and Guided Self-finetuning.

### 4.1. Decoupled State Diffusion (DSD)

**Decoupling Control Estimation using Inverse Dynamics.** There are deep connections behind states, controls, and constraints, considering both the relationship between control and state and learning the dynamics behind state evolution. Such relation becomes more complex as the dimension of the system states goes up. However, some diffusion-based methods (e.g., DiffPhyCon (Wei et al., 2024)), which jointly diffuse over state and input, learn the relationship implicitly and may generate physically inconsistent state-control pairs that violate the underlying system dynamics. Additionally, control actions are less smooth than states, making their distribution more challenging to model (Ajay et al., 2022).

Therefore, rather than jointly sampling both control signals and intermediate states using the denoising network, we choose to decouple them and diffuse only states $\mathbf{y}$, i.e.

$$\mathbf{x} := \mathbf{y}_{[0:T]}.$$

Then we update the prediction of control $\mathbf{u}$ sequence by inputting the generated state trajectory to an inverse dynamic model $f_\phi$:

$$\mathbf{u}_{t,\text{update}}^0 = f_\phi(\mathbf{y}_t^0, \mathbf{y}_{t+1}^0),$$

where $0$ denotes the final output of the denoising timestep from the diffusion model. We parameterize it using an Autoregressive MLP and optimize it simultaneously with the denoiser via training data. Our final optimization loss function is:

$$\begin{aligned} L(\theta, \phi) &:= \mathbb{E}_{\mathbf{x},k,\mathbf{y}_0^*,\mathbf{y}_f,\epsilon}[||\mathbf{x} - \mathbf{x}_\theta(\mathbf{x}^k, k, \mathbf{y}_0^*, \mathbf{y}_f)||^2] \\ &+ \mathbb{E}_{\mathbf{y}_t,\mathbf{u}_t,\mathbf{y}_{t+1}}[||\mathbf{u}_t - f_\phi(\mathbf{y}_t, \mathbf{y}_{t+1})||^2], \end{aligned} \quad (2)$$

where $\mathbf{y}_t, \mathbf{u}_t, \mathbf{y}_{t+1}$ are sampled from the dataset. Note that the data used to train the diffusion model can also be utilized to train $f_\phi$.

**Cost Optimization via Gradient Guidance.** After training both models, we optimize the cost function $J$ through inference-time gradient guidance. During the denoising process, we modify the sampling procedure by incorporating cost gradients:

$$\begin{aligned} \mu_\theta(\mathbf{x}^k, k, \mathbf{y}_0^*, \mathbf{y}_f) &= \frac{\sqrt{\bar{\alpha}^{k-1}}\beta^k}{1-\bar{\alpha}^k}\hat{\mathbf{x}}^0 + \frac{\sqrt{\alpha^k}(1-\bar{\alpha}^{k-1})}{1-\bar{\alpha}^k}\mathbf{x}^k \\ &- \lambda\Sigma^k\nabla_{\mathbf{x}^k}J(\hat{\mathbf{x}}^0(\mathbf{x}^k)), \end{aligned} \quad (3)$$

where $\lambda$ controls guidance strength, $\Sigma^k$ is the noise scale at step $k$, $\alpha^k := 1 - \beta^k$ and $\bar{\alpha}^k := \prod_{s=1}^k \alpha^s$. Since our diffusion model operates on states only, we recover control inputs using the inverse dynamics model $f_\phi$ at each step. This approach enables optimization of arbitrary cost functions without model retraining, while maintaining trajectory feasibility through the learned diffusion process.

**Target-conditioning as Inpainting.** Modeling whether the generated trajectory accurately satisfies the initial state of $\mathbf{y}_0^*$ and the desired target state of $\mathbf{y}_f$ can also be regarded as a constraint satisfaction of equations, that is, the generated trajectory should contain the start and target. We adopt a more direct method to solve this: we not only input it as an additional condition to the diffusion denoising network but also treat it as an inpainting problem similar to image generation. In brief, we substitute the corresponding location in the sampled trajectories $\mathbf{x}^{k-1} \sim p_\theta(\mathbf{x}^{k-1}|\mathbf{x}^k, \mathbf{y}_0^*, \mathbf{y}_f)$ with the given start and target $\mathbf{y}_0^*, \mathbf{y}_f$ after all diffusion timesteps, analogously to observed pixels in image generation (Lugmayr et al., 2022).

### 4.2. Dual Mode Decomposition (DMD) for Denoising Module

In this section, we propose a design for the denoising network that decomposes the modeling of linear and nonlinear modes in the sampled trajectory by a dual-Unet architecture, as shown in Figure 1.

Our design draws inspiration from control theory. For linear systems, Yan et al. (2012) demonstrated that optimal control signals have a linear relationship with a specific linear combination $\mathbf{y}_c$ of initial and target states. Building upon this insight, we develop a framework where a bias-free linear layer first learns this crucial linear combination $\mathbf{y}_c$ from the initial state $\mathbf{y}_0$ and target state $\mathbf{y}_f$. The first UNet then learns coefficients that map $\mathbf{y}_c$ to control signals, establishing first-order terms, while the second UNet learns coefficients for quadratic terms. These quadratic terms, introduced through residual connections, refine the first-order approximation and enhance the network's capacity to model complex dynamics. Note that the nonlinear terms essentially come from the nonlinearity of the dynamics.

The implementation includes several key components. The network accepts as input: (1) noisy trajectory $\mathbf{x}^k$ with dimension ($N$) corresponding to the network's channel dimension; (2) initial state $\mathbf{y}_0$ and target state $\mathbf{y}_f$, which generate $\mathbf{y}_c$ through a bias-less linear layer; and (3) diffusion timesteps $k$, encoded via sinusoidal embedding (Ho et al., 2020) as $\mathbf{k}_{\text{emb}}$. The first UNet generates first-order coefficients to compute an initial prediction using $\mathbf{y}_c$. Subsequently, the second UNet combines these features to generate quadratic coefficients, producing correction terms through quadratic operations with $\mathbf{y}_c$. The final output $\hat{x}^0$ combines these components to predict the denoised trajectory.

The architecture decomposes system dynamics into linear and nonlinear components, effectively handling complex features in nonlinear control systems while maintaining numerical stability during training. The first-order and quadratic terms based on $\mathbf{y}_c$ incorporate fundamental control principles into the network structure, providing effective constraints for the learning process. This structured inductive bias significantly improves the model's data efficiency, enabling reliable control strategy learning from limited training samples.

The network performs sequential transformations on the input signals. Let $B$ denote batch size, $T$ sequence length, $C_1$ and $C_2$ feature dimensions, and $N$ the dimension of $\mathbf{y}_c$. The input noisy trajectory $\mathbf{x}^k \in \mathbb{R}^{B \times T \times N}$ and $\mathbf{y}_c \in \mathbb{R}^{B \times C_1}$ are processed through two UNets to generate first-order and quadratic predictions:

$$\mathbf{C}_1 = \text{UNet1}(\mathbf{x}^k, \text{kemb}), \quad (4)$$

$$\mathbf{O}_1 = \text{reshape}(\mathbf{C}_1) \cdot \mathbf{y}_c, \quad (5)$$

$$\mathbf{C}_2 = \text{UNet}_2([\mathbf{x}^k, \text{C1}], \text{kemb}), \quad (6)$$

$$\mathbf{O}_2 = \mathbf{y}_c^T \cdot \text{reshape}(\mathbf{C}_2) \cdot \mathbf{y}_c, \quad (7)$$

$$\hat{\mathbf{x}}^0 = \mathbf{O}_1 + \mathbf{O}_2, \quad (8)$$

where $\mathbf{C}_1$ produces first-order coefficients $\in \mathbb{R}^{B \times T \times (N \times C_1)}$, $\mathbf{O}_1$ computes linear predictions $\in \mathbb{R}^{B \times T \times N}$, $\mathbf{C}_2$ generates quadratic coefficients $\in \mathbb{R}^{B \times T \times (C_1 \times N \times C_1)}$, and both $\mathbf{O}_2$ and the final output $\hat{\mathbf{x}}^0$ are $\in \mathbb{R}^{B \times T \times N}$. We illustrate the structural framework of the denoising network in Figure 1.

### 4.3. Guided Self-finetuning (GSF)

Randomly generated training data cannot guarantee coverage of optimal scenarios. To generate near-optimal controls that may deviate significantly from the training distribution. To address this limitation, we propose leveraging the model's initially generated data (under the guidance of cost function), which naturally deviates from the training distribution toward optimality, for iterative retraining to systematically expand the exploration space. This approach maintains physical consistency by ensuring generated samples adhere to the underlying system dynamics.

Our methodology involves extracting control sequences from the generated samples (i.e., the output of inverse dynamics $\mathbf{u}^0_{\text{update}}$) and reintroduces it into the system to interact and generate corresponding state sequences $\mathbf{y}^0_{\text{update}}$. Together we add the renewed $[\mathbf{u}^0_{\text{update}}, \mathbf{y}^0_{\text{update}}]$ to the retrain data pool used for a new round of fine-tuning, notably without requiring explicit system parameter identification. We iterate this process over multiple rounds specified by a hyperparameter, systematically expanding the model's exploration space to progressively approach optimal control policy. Denote the sampling process under cost $J$'s guidance and the following interacting process as $[\mathbf{u}^0_{\text{update}}, \mathbf{y}^0_{\text{update}}] = \mathcal{S}(\mathbf{x}^K, \mathbf{y}^*_0, \mathbf{y}_f, J, \Phi)$. The process can be formulated as:

$$[\mathbf{u}^0_{\text{update}}, \mathbf{y}^0_{\text{update}}] = \mathcal{S}_{(\mathbf{x}^K, \mathbf{y}^*_0, \mathbf{y}_f) \sim D}(\mathbf{x}^K, \mathbf{y}^*_0, \mathbf{y}_f, J, \Phi), \quad (9)$$

$$D = [D, [\mathbf{u}^0_{\text{update}}, \mathbf{y}^0_{\text{update}}]], \quad (10)$$

where $D$ is the training set.

We provide the algorithm form of SEDC in Appendix 1.

## 5. Experiments

**Experiment settings.** We conducted experiments on three nonlinear systems, following the instructions in the previous works for data synthesis. These systems include: the 1-D Burgers dynamics (Hwang et al., 2022; Wei et al., 2024), which serves as a fundamental model for studying nonlinear wave propagation and turbulent fluid flow; the

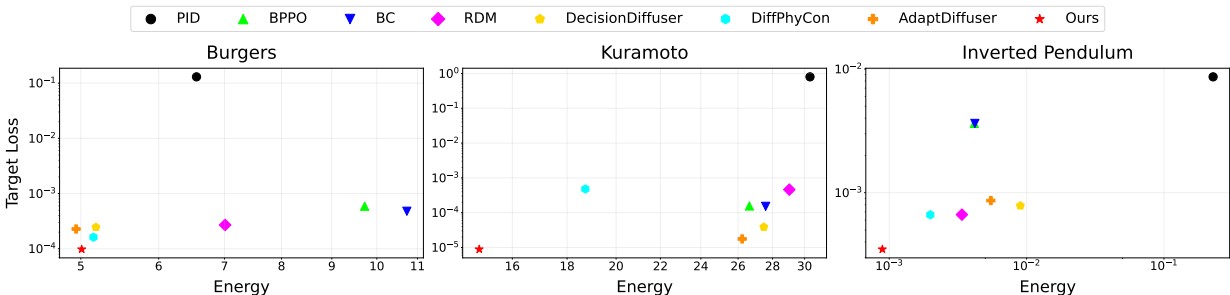

*Figure 2.* Comparison of target loss and energy cost $J$ across different datasets. The closer the data point is to the bottom left, the better the performance.

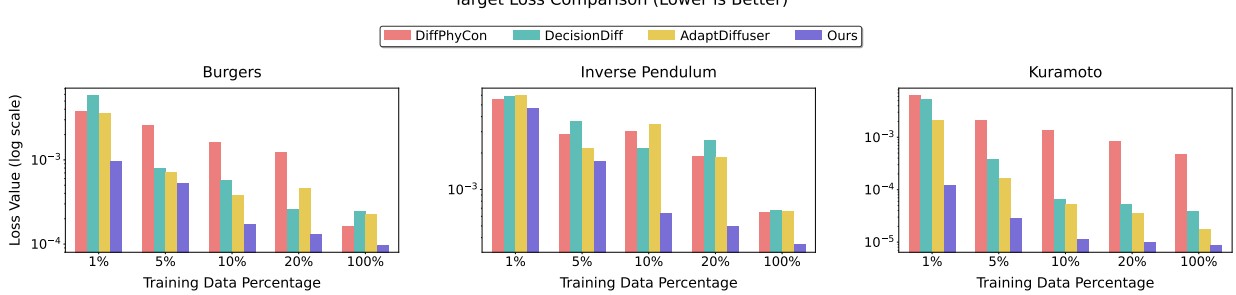

*Figure 3.* Sample-efficiency comparison on Burgers, Kuramoto and Inverse Pendulum dynamics.

Kuramoto dynamics (Acebrón et al., 2005; Baggio et al., 2021; Gupta et al., 2022), which is essential for understanding synchronization phenomena in complex networks and coupled oscillator systems; and the inverted pendulum dynamics (Boubaker, 2013), which represents a classical benchmark problem in nonlinear control theory and robotic systems. For each system, we generated control/state trajectory data using the finite difference method and selected 50 trajectories as the test set. Detailed descriptions of the system dynamics equations and data synthesis procedures are provided in the appendix.

We evaluate two metrics which is crucial in complex system control: **Target Loss**, the mean-squared-error (MSE) of $\mathbf{y}_T$ and desired target $\mathbf{y}_f$, i.e. $\frac{1}{N}\|\mathbf{y}_T - \mathbf{y}_f\|^2$. (Note that $\mathbf{y}_T$ is obtained by simulating the real system using the control inputs generated by each method, along with the given initial state conditions, rather than extracted from the sample trajectories of the diffusion-based methods); **Energy** $J = \int_0^T |\mathbf{u}(t')|^2 dt'$, which measures the cumulative control effort required to achieve the target state. Lower values of both metrics indicate better performance.

**Baselines.** We select the following state-of-the-art(SOTA) baseline methods for comparison. For traditional control approaches, we employ the classical PID (Proportional-Integral-Derivative) controller (Li et al., 2006), which remains widely used in industrial applications. For supervised learning ,we employ Behavioral Cloning (BC) (Pomerleau, 1988), an established imitation learning approach. In

terms of reinforcement learning methods, we incorporate BPPO (Zhuang et al., 2023), a state-of-the-art algorithm. For diffusion-based methods, we include several recent prominent approaches: DecisionDiffuser (DecisionDiff) (Ajay et al., 2022), which is a SOTA classifier-free diffusion-based planner; AdaptDiffuser (Liang et al., 2023), which enhances DecisionDiffuser with a self-evolving mechanism; RDM (Zhou et al., 2024), which adaptively determines the timing of complete control sequence sampling; and DiffPhyCon (Wei et al., 2024), which is specifically designed for controlling complex physical systems. Detailed descriptions of the baselines are included in Appendix D.

### 5.1. Overall Control Performance

**Results.** In Figure 2, we compare different methods' performance across three dynamical systems using two-dimensional coordinate plots, where proximity to the lower-left corner indicates better trade-offs between control accuracy and energy efficiency. Since unstable control can lead to system failure regardless of energy efficiency, we prioritize control accuracy and report metrics at each method's minimum Target Loss. Our method achieves the closest position to the lower-left corner in three datasets, demonstrating best balance between accuracy and efficiency despite varying system characteristics. We achieve the best Target Loss across all systems, outperforming the best baselines by 39.5%, 49.4%, and 47.3% in Burgers, Kuramoto, and IP systems respectively. This superior performance reflects

our method's enhanced dynamics learning capability under identical sampling conditions. For energy efficiency, our method leads in Kuramoto and IP systems while remaining competitive in Burgers, trailing AdaptDiffuser by only 1.3%.

Regarding method types, traditional PID control shows the poorest performance, as system complexity exacerbates the difficulties in PID control and tuning. RL-based methods are competitive against some diffusion- but sacrifice Target Loss performance and underperform compared to Diffusion-based methods in other systems. Diffusion-based methods demonstrate superior overall performance, as they better capture long-term dependencies in system dynamics compared to traditional and RL methods, avoiding myopic failure modes and facilitating global optimization of long-term dynamics.

Due to the space limits, we present detailed numeric results in Appendix E. Moreover, we visualize the control dynamics of SEDC and SOTA baselines in Appendix G.

### 5.2. Sample Efficiency

**Experiment settings.** To evaluate the sample efficiency of diffusion-based methods, we conducted experiments on all the systems using varying proportions of the full training dataset. Specifically, we trained models using 1%, 5%, 10%, 20%, and 100% of the available data and assessed their performance using the Target Loss metric on a held-out test set.

**Results.** Figure 3 demonstrates our method's superior performance in controlling Burgers and Kuramoto systems compared to state-of-the-art baselines. In all systems, our approach achieves significantly lower target loss values across all training data percentages. Most notably, with only 10% of the training data, our method attains a target loss of 1.71e-4 for Burgers, 1.12e-5 for Kuramoto, and 6.35e-4 for Inverse Pendulum, matching(-5.5% in Burgers) or exceeding(+36.4% in Kuramoto abd +1.2% in Inverse Pendulum) the performance of best baseline methods trained on the complete dataset. This indicates our method can achieve state-of-the-art performance while requiring only 10% of the training samples.

### 5.3. Ablation Study

**Overall ablation study.** We explore the main performance against each ablation of the original SEDC. Specifically, *w/o DSD* removes the inverse dynamics, unifying the diffusion of system state and control input, i.e. $\mathbf{x} = [\mathbf{u}, \mathbf{y}]$. Therefore, the diffusion model is required to simultaneously capture the temporal information and implicit dynamics of the control and system trajectory. Note that the inpainting mechanism and gradient guidance are retained. *w/o DMD* removes the

*Table 1.* Performance comparison of different ablations across multiple datasets. **Target loss** results with 10% and 100% training sample for each method are reported. The best, second-best and worst results of each row are highlighted in **bold**, underlined and *italics*, respectively.

| System | Ratio | Ours | Ours/DSD | Ours/DMD | Ours/GSF |
|---|---|---|---|---|---|
| Burgers | 10% | **1.74e-4** | *1.00e-3* | 3.78e-4 | 6.67e-4 |
| | 100% | **9.80e-5** | *8.71e-4* | 2.28e-4 | 2.62e-4 |
| Kuramoto | 10% | **1.12e-5** | *4.15e-3* | 5.21e-5 | 4.77e-5 |
| | 100% | **8.90e-6** | *5.43e-3* | 1.76e-5 | 3.88e-5 |
| IP | 10% | **6.21e-4** | *1.58e-3* | 1.10e-3 | 2.00e-3 |
| | 100% | **3.49e-4** | *1.37e-3* | 6.64e-4 | 7.85e-4 |

*Table 2.* SEDC and *w/o DSD* Target Loss comparison across different state dimensions in Kuramoto. The Dec. indicates the reduction in target loss achieved by SEDC compared to *w/o DSD*. Notably, the magnitude of loss reduction increases proportionally with network dimensionality, demonstrating that DSD's performance enhancement scales positively with dimensional growth.

| $N$ | **4** | **5** | **6** | **7** | **8** |
|---|---|---|---|---|---|
| SEDC (e-6) | 3.45 | 2.89 | 5.67 | 4.12 | 8.90 |
| w/o DSD (e-4) | 3.98 | 3.23 | 16.78 | 12.45 | 54.32 |
| Dec.(%) | 99.13 | 99.11 | 99.66 | 99.67 | 99.84 |

decomposition design, resulting in a single 1-D Unet structure as the denoising network, following DecisionDiff (Ajay et al., 2022). Finally, *w/o GSF* reports the performance without iterative self-finetuning, which means the model only uses the original dataset to train itself. To show the sample-efficiency performance, we also investigate the results under less amount of training sample(10%). For *w/o DMD* and *w/o DSD*, we adjust the number of trainable parameters at a comparable level against the original version.

Table 1 shows the Target Loss performance of different ablations of SEDC across multiple datasets and different training sample ratios. As can be seen, removing any component leads to a certain decrease in performance, whether the training data is limited or not, demonstrating the effectiveness of each design. The most significant performance drops are often observed in *w/o DSD*, highlighting the importance of explicit learning of dynamics in complex systems. *w/o DMD* exhibits the lowest decline across the three systems. This is because the single-Unet-structured denoising network can already capture the nonlinearity to some extent, but not as good as the proposed decomposition approach. with 10% of training data, removing individual components still led to noticeable performance degradation, and the patterns consistent with the full dataset results. This demonstrates that our designs remain effective in low-data scenarios.

**Effectiveness of DSD.** To evaluate DSD's effectiveness against the curse of dimensionality, we compared the per-

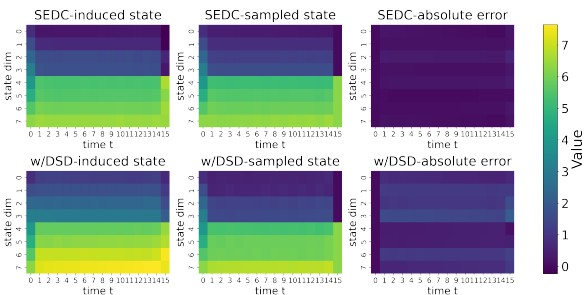

*Figure 4.* Comparison of State Trajectory Consistency between SEDC and w/o DSD Models. The heatmaps show induced states (left), sampled states (middle), and their absolute differences (right) for both SEDC (top) and w/o DSD (bottom) approaches under identical start-target conditions.

*Table 3.* Performance degradation using different denoiser output with varying nonlinearity strength $\gamma$ in the Kuramoto system. The Dec. indicates the reduction in target loss achieved by nonlinear output $\mathbf{O}_1 + \mathbf{O}_2$ compared to linear output $\mathbf{O}_1$. Notably, the magnitude of loss reduction increases proportionally with the nonlinearity strength $\gamma$, indicating that the quadratic term exhibits enhanced capability in capturing nonlinear dynamics as the system's nonlinearity intensifies.

| $\gamma$ | 1 | 2 | 4 |
|---|---|---|---|
| $\mathbf{O}_1 + \mathbf{O}_2$ | 8.90e-6 | 2.78e-5 | 3.89e-5 |
| $\mathbf{O}_1$ | 1.42e-5 | 4.73e-5 | 8.52e-5 |
| Dec. (%) | 37.3 | 41.2 | 54.3 |

formance of original and *w/o DSD* models across Kuramoto systems with dimensions ranging from $N = 4$ to $N = 8$. Experimental results (Table 2) show that performance degradation (Dec.) from *w/o DSD* increases with system dimensionality, demonstrating DSD's enhanced effectiveness in higher-dimensional systems and validating its capability to address dimensionality challenges.

To investigate the effectiveness of dynamical learning, we compared the consistency between action sequences and diffusion-sampled state trajectories in models with and without DSD. While both approaches can sample state trajectories from diffusion samples, they differ in action generation: SEDC uses inverse dynamics prediction, whereas *w/o DSD* obtains actions directly from diffusion samples by simultaneously diffusing states and control inputs. We test both models using identical start-target conditions and visualize the state induced from the generated actions and the state sampled from the diffusion model, along with the difference (error) between the above two states in Figure 4. We can observe that SEDC's action-induced state trajectories showed significantly higher consistency with sampled trajectories compared to *w/o DSD*, demonstrating that DSD using inverse dynamics achieves more accurate learning of control-state dynamical relationships.

**Effectiveness of DMD.** To investigate the contribution of DMD's dual-Unet architecture to nonlinearity learning, we

conducted experiments on the Kuramoto system with varying degrees of nonlinearity (controlled by the coefficient $\gamma \in \{1, 2, 4\}$ of the nonlinear sinusoidal term, where larger values indicate stronger nonlinearity). We compared the performance between using only the linear intermediate output ($\hat{\mathbf{x}}_0 = \mathbf{O}_1$) of the denoising network and the original nonlinear output ($\hat{\mathbf{x}}_0 = \mathbf{O}_1 + \mathbf{O}_2$) in terms of Target Loss. As shown in Table 3, the performance degradation (Dec.) from using only $\mathbf{O}_1$ becomes more pronounced as nonlinearity increases. This demonstrates both the significance of the nonlinear branch $\mathbf{O}_2$ in capturing strong nonlinear dynamics and the effectiveness of decoupling linear and nonlinear modes in handling system nonlinearity.

**Effectiveness of GSF.** To validate GSF's effectiveness in guiding the model toward learning the optimal (energy-efficient) target distribution, we conduct ablation studies on fine-tuning rounds and evaluate control signal energy on the test set. Results show decreasing energy metrics over rounds, confirming our approach's convergence behavior toward near-optimal control strategies. Due to space limits, results and detailed discussion are provided in Appendix F.

## 6. Conclusion

In this paper, we presented SEDC, a novel sample-efficient diffusion-based framework for complex nonlinear system control. By addressing fundamental challenges in data-driven control through three key innovations - Decoupled State Diffusion (DSD), Dual-Mode Decomposition (DMD), and Guided Self-finetuning (GSF) - SEDC achieves superior control performance while significantly reducing sample requirements. Our comprehensive experiments across three nonlinear systems demonstrate that SEDC outperforms existing methods by 39.5%-49.4% in control accuracy while maintaining computational efficiency. Most notably, SEDC achieves state-of-the-art performance using only 10% of the training samples required by baseline methods, marking a significant advancement in sample-efficient control of complex systems. These results validate our approach's effectiveness in addressing the curse of dimensionality, handling strong nonlinearities, and bridging the gap between non-optimal training data and optimal control solutions. As complex system control continues to evolve across various domains, SEDC's sample-efficient framework provides a promising direction for future research and practical applications in data-driven control.

## Impact Statement

This paper presents work whose goal is to advance the field of Machine Learning. There are many potential societal consequences of our work, none which we feel must be specifically highlighted here.

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

# A. Algorithm form of SEDC

---

**Algorithm 1** SEDC: Training and finetuning

---

**Input:** Initial dataset $\mathcal{D}_0$, diffusion steps $K$, guidance strength $\lambda$, self-finetuning rounds $R$, forward dynamics $f_{\text{forward}}$
**Output:** Optimized trajectory $\mathbf{y}_{0:T}^0$, controls $\mathbf{u}_{0:T}^0$
**Function** *Initial Training($\mathcal{D}_0$)*
> **while** *not converged* **do**
>> Sample batch $(\mathbf{y}_{0:T}, \mathbf{u}_{0:T}) \sim \mathcal{D}_0$
>> Sample $k \sim \mathcal{U}\{1, ..., K\}$, $\epsilon \sim \mathcal{N}(0, I)$
>> Corrupt states: $\mathbf{y}^k = \sqrt{\bar{\alpha}^k}\mathbf{y} + \sqrt{1 - \bar{\alpha}^k}\epsilon$
>> Predict clean states: $\hat{\mathbf{y}}^0 = \mathcal{G}_\theta(\mathbf{y}^k, k, \mathbf{y}_0^*, \mathbf{y}_f)$
>> Predict controls: $\hat{\mathbf{u}}_t = f_\phi(\hat{\mathbf{y}}_t^0, \hat{\mathbf{y}}_{t+1}^0)$
>> Compute losses: $\quad L_{\text{diff}} = \|\mathbf{y} - \hat{\mathbf{y}}^0\|^2$
>> $L_{\text{inv}} = \|\mathbf{u}_t - \hat{\mathbf{u}}_t\|^2$
>> Update $\theta, \phi$ with $\nabla(L_{\text{diff}} + L_{\text{inv}})$

**for** $r = 1$ **to** $R$ **do**
> **Guided Data Generation:**
> Initialize $\mathbf{y}^K \sim \mathcal{N}(0, I)$, sample $(\mathbf{y}_0^*, \mathbf{y}_f) \sim \mathcal{D}_{r-1}$
> **for** $k = K$ **downto** 1 **do**
>> Predict $\hat{\mathbf{y}}^0 = \mathcal{G}_\theta(\mathbf{y}^k, k, \mathbf{y}_0^*, \mathbf{y}_f)$
>> Compute gradient: $g = \nabla_{\mathbf{y}^k} J(\hat{\mathbf{y}}^0)$
>> Adjust mean: $\mu_\theta = \mu_\theta^{(\text{base})} - \lambda\Sigma^k g$
>> Sample $\mathbf{y}^{k-1} \sim \mathcal{N}(\mu_\theta, \Sigma^k I)$
>> Enforce constraints: $\mathbf{y}^{k-1}[0] \leftarrow \mathbf{y}_0^*, \mathbf{y}^{k-1}[T] \leftarrow \mathbf{y}_f$
>
> Recover controls: $\mathbf{u}_t^0 = f_\phi(\mathbf{y}_t^0, \mathbf{y}_{t+1}^0)$
> **System Interaction:**
> Generate $\mathbf{y}_{\text{update}}^0 = f_{\text{forward}}(\mathbf{u}_{0:T}^0, \mathbf{y}_0^*)$
> Augment dataset: $\mathcal{D}_r = \mathcal{D}_{r-1} \cup \{(\mathbf{y}_{\text{update}}^0, \mathbf{u}_{0:T}^0)\}$
>
> **Adaptive Fine-tuning:**
> **while** *validation loss decreases* **do**
>> Sample batch from $\mathcal{D}_r$
>> Perform training steps as in Initial Training

**return** Optimized $\theta, \phi$
*(Test process follows guided data generation with test conditions $(\mathbf{y}_0^*, \mathbf{y}_f)$ provided.)*

---

# B. Detailed System and Dataset Description

## B.1. Burgers Dynamics

The Burgers' equation is a governing law occurring in various physical systems. We consider the 1D Burgers' equation with the Dirichlet boundary condition and external control input $\mathbf{u}(t, x)$:

$$\begin{cases} \frac{\partial y}{\partial t} = -y \cdot \frac{\partial y}{\partial x} + \nu \frac{\partial^2 y}{\partial x^2} + \mathbf{u}(t, x) & \text{in } [0, T] \times \Omega \\ y(t, x) = 0 & \text{on } [0, T] \times \partial\Omega \\ y(0, x) = y_0(x) & \text{in } \{t = 0\} \times \Omega \end{cases}$$

Here $\nu$ is the viscosity parameter, and $y_0(\mathbf{x})$ is the initial condition. Subject to these equations, given a target state $y_d(x)$, the objective of control is to minimize the control error $\mathcal{J}_{\text{actual}}$ between $y_T$ and $y_d$, while constraining the energy cost $\mathcal{J}_{\text{energy}}$ of the control sequence $\mathbf{u}(t, x)$.

We follow instructions in (Wei et al., 2024) to generate a 1D Burgers' equation dataset. Specifically, for numerical simulation,

we discretized the spatial domain [0,1] and temporal domain [0,1] using the finite difference method (FDM). The spatial grid consisted of 128 points, while the temporal domain was divided into 10000 timesteps. We initiated the system with randomly sampled initial conditions and control inputs drawn from specified probability distributions. This setup allowed us to generate 90000 trajectories for training and 50 trajectories for testing purposes.

### B.2. Kuramoto Dynamics

The Kuramoto model is a paradigmatic system for studying synchronization phenomena. We considered a ring network of $N = 8$ Kuramoto oscillators. The dynamics of the phases (states) of oscillators are expressed by:

$$\dot{\theta}_{i,t} = \omega + \gamma(\sin(\theta_{i-1,t-1} - \theta_{i,t-1}) + \sin(\theta_{i+1,t-1} - \theta_{i,t-1})) + u_{i,t-1}, \quad i = 1, 2, ..., N. \tag{11}$$

For the Kuramoto model, we generated 20,000 samples for training and 50 samples for testing. The initial phases were sampled from a Gaussian distribution $\mathcal{N}(0, I)$, and the random intervention control signals were sampled from $\mathcal{N}(0, 2I)$. The system was simulated for $T = 16$ time steps with $\omega = 0$, following Baggio et al. (2021). The resulting phase observations and control signals were used as the training and test datasets.

### B.3. Inverted Pendulum Dynamics

The inverted pendulum is a classic nonlinear control system. The dynamics can be represented by:

$$\frac{d^2\theta}{dt^2} = \frac{g}{L}\sin(\theta) - \frac{\mu}{L}\frac{d\theta}{dt} + \frac{1}{mL^2}u$$

where $\theta$ is the angle from the upward position, and $u$ is the control input torque. The system parameters are set as: gravity $g = 9.81$ m/s², pendulum length $L = 1.0$ m, mass $m = 1.0$ kg, and friction coefficient $\mu = 0.1$.

To generate the training dataset, we simulate 90,000 trajectories for training and 50 for testing with 128 time steps each, using a time step of 0.01s. For each trajectory, we randomly sample initial states near the unstable equilibrium point with $\theta_0 \sim \mathcal{U}(-1, 1)$ and $\dot{\theta}_0 \sim \mathcal{U}(-1, 1)$, and generate control inputs from $u \sim \mathcal{U}(-0.5, 0.5)$. The resulting dataset contains the state trajectories and their corresponding control sequences.

## C. Implementation Details

### C.1. Implementation of SEDC

In this section, we describe various architectural and hyperparameter details:

- The temporal U-Net (1D-Unet) (Janner et al., 2022) in the denoising network consists of a U-Net structure with 4 repeated residual blocks. Each block comprises two temporal convolutions, followed by group normalization, and a final Mish nonlinearity. The channel dimensions of the downsample layers are $1, 2, 4 * statedimension$. Timestep embedding is produced by a Sinusoidal Positional Encoder, following a 2-layer MLP, and the dimension of this embedding is 32. The dimension of condition embedding is the same as the system state dimension.

- We represent the inverse dynamics $f_\phi$ with an autoregressive model with 64 hidden units and ReLU activations. The model autoregressively generates control outputs along the control dimensions.

- We train $\mathbf{x}_\theta$ and $f_\phi$ using the Adam optimizer with learning rates from {1e-3, 5e-3, 1e-4}. The exact choice varies by task. Moreover, we also use a learning rate scheduler with step factor=0.1. Training batch size is 32.

- We use $K = 128$ diffusion steps.

- We use a guidance scale $\lambda \in \{0.01, 0.001, 0.1\}$ but the exact choice varies by task.

Table 4. Approximate Training Time Comparison of Different Models on Various Datasets (in hours)

| Dataset/System | DecisionDiffuser | RDM | DiffPhyCon | AdaptDiffuser | SEDC |
|---|---|---|---|---|---|
| Burgers | 2.5 | 2.5 | 3.0 | 2.5 | 2.5 |
| Kuramoto | 1.5 | 1.5 | 1.5 | 1.0 | 1.0 |
| IP | 1.0 | 1.0 | 1.5 | 1.0 | 0.5 |
| Swing | 1.5 | 1.5 | 2.0 | 1.5 | 1.0 |

Table 5. Approximate Inference Time Comparison of Different Models on Various Datasets (in seconds)

| Dataset/System | DecisionDiffuser | RDM | DiffPhyCon | AdaptDiffuser | SEDC |
|---|---|---|---|---|---|
| Burgers | 3.0 | 4.0 | 6.0 | 4.0 | 4.0 |
| Kuramoto | 1.0 | 1.5 | 2.0 | 1.5 | 1.5 |
| IP | 0.5 | 1.0 | 1.0 | 0.5 | 0.5 |
| Swing | 1.5 | 2.0 | 2.5 | 1.5 | 1.5 |

### C.2. Training and Inference Time Analysis

The diffusion-based methods are trained on single NVIDIA GeForce RTX 4090 GPU. We evaluate the training and inference time of all the diffusion-based methods evaluated in the experiment session. As shown in Table 4, we compare the training efficiency of different models across various datasets. DiffPhyCon consistently shows longer training times compared to other methods, because it requires training two models that learn the joint distribution and the prior distribution respectively, increasing its training time consumption. The training times of DecisionDiffuser, RDM, and AdaptDiffuser are generally comparable, while SEDC demonstrates relatively efficient training performance across most datasets. This may be because of the proposed designs that not only improve sample efficiency but also improve learning efficiency.

The inference time comparison in Table 5 reveals that DiffPhyCon requires longer execution time compared to other models, because it needs to sample from two learned distributions in the denoising process. RDM achieves relatively slower inference speeds than DecisionDiffuser, AdaptDiffuser, and SEDC, because RDM replans during inference, increasing planning time. Notably, all models exhibit shorter training and inference times on the IP dataset, suggesting the influence of system complexity on computational efficiency.

## D. Baselines Description

### D.1. PID

PID (Proportional-Integral-Derivative) control is a classical feedback control methodology that has been widely adopted in industrial applications. The control signal is generated by computing the weighted sum of proportional, integral, and derivative terms of the error. The control law can be expressed as:

$$u(t) = K_p e(t) + K_i \int_0^t e(\tau)d\tau + K_d \frac{d}{dt}e(t)$$

While PID controllers exhibit robust performance and require minimal system modeling, their effectiveness may be compromised when dealing with highly nonlinear or time-varying systems, necessitating frequent parameter tuning.

### D.2. BC, BPPO

Behavior Cloning (BC) represents a supervised imitation learning paradigm that aims to learn a direct mapping from states to actions by minimizing the deviation between predicted actions and expert demonstrations. Despite its implementation simplicity and sample efficiency, BC suffers from distributional shift, where performance degradation occurs when encountering states outside the training distribution. The objective function can be formulated as:

$$L_{BC}(\theta) = \mathbb{E}_{(s,a)\sim\mathcal{D}}[-\log \pi_\theta(a|s)]$$

where $\mathcal{D}$ denotes the expert demonstration dataset.

Behavior-guided PPO (BPPO) presents a hybrid approach that integrates behavior cloning with Proximal Policy Optimization. By incorporating a behavioral cloning loss term into the PPO objective, BPPO facilitates more efficient policy learning while maintaining the exploration capabilities inherent to PPO. The composite objective function is defined as:

$$L_{BPPO}(\theta) = L_{PPO}(\theta) + \alpha L_{BC}(\theta)$$

where $\alpha$ serves as a balancing coefficient between the PPO and BC objectives.

Each method exhibits distinct characteristics: BC demonstrates effectiveness when abundant high-quality expert demonstrations are available. BPPO leverages the synergy between expert knowledge and reinforcement learning for complex control scenarios.

### D.3. Diffusion-based methods

- **DecisionDiffuser:**
  A novel approach that reformulates sequential decision-making as a conditional generative modeling problem rather than a reinforcement learning task. The core methodology involves modeling policies as return-conditional diffusion models, enabling direct learning from offline data without dynamic programming. The model can be conditioned on various factors including constraints and skills during training.

- **DiffPhyCon:**
  A diffusion-based method for controlling physical systems that operates by jointly optimizing a learned generative energy function and predefined control objectives across entire trajectories. The approach incorporates a prior reweighting mechanism to enable exploration beyond the training distribution, allowing the discovery of diverse control sequences while respecting system dynamics.

- **AdaptDiffuser:**
  An evolutionary planning framework that enhances diffusion models through self-evolution. The method generates synthetic expert data using reward gradient guidance for goal-conditioned tasks, and employs a discriminator-based selection mechanism to identify high-quality data for model fine-tuning. This approach enables adaptation to both seen and unseen tasks through continuous model improvement.

- **RDM:**
  A replanning framework for diffusion-based planning systems that determines replanning timing based on the diffusion model's likelihood estimates of existing plans. The method introduces a mechanism to replan existing trajectories while maintaining consistency with original goal states, enabling efficient bootstrapping from previously generated plans while adapting to dynamic environments.

## E. Numeric Results of Figure 2

We leverage 2-D plots in the main paper to better illustrate the performance comparison of all the methods. Here we provide the provides the corresponding numerical results in detail in Table 6.

## F. Results and Discussion of the ablation study on GSF

To validate GSF's effectiveness in guiding the model toward learning the optimal (energy- efficient) target distribution, we conduct ablation studies on fine-tuning rounds and test control signal energy on the test set. The result is provided in Table 7. Before finetuning, energy performance is the poorest because of the non-optimality of the initial training samples, and the first round of GSF greatly supplemented the training toward optimality, as the energy of the produced control inputs decreases at an average of 38.4%. The increases observed in the second round persist but are significantly lower than those in the first round. This may be because the first round of GSF has already captured the most significant deviations toward

*Table 6.* **Performance comparison of different models across three datasets.** Lower values indicate better performance for both metrics. Best and second-best results of each row are highlighted in **bold** and underlined respectively.

| Model | Burgers | | Kuramoto | | IP | |
|---|---|---|---|---|---|---|
| | Target Loss | $J$(Energy) | Target Loss | $J$(Energy) | Target Loss | $J$(Energy) |
| PID | 1.30e-1 | 6.56 | 7.99e-1 | 30.35 | 8.64e-3 | 2.28e-1 |
| BPPO | 5.90e-4 | 9.72 | 1.56e-4 | 26.64 | 3.63e-3 | 4.16e-3 |
| BC | 4.78e-4 | 10.73 | 1.52e-4 | 27.59 | 3.63e-3 | 4.20e-3 |
| DecisionDiffuser | 2.46e-4 | 5.18 | 3.88e-5 | 27.48 | 6.65e-4 | 9.00e-4 |
| RDM | 2.70e-4 | 7.01 | 4.60e-4 | 29.03 | 7.85e-4 | 3.38e-3 |
| DiffPhyCon | 1.62e-4 | 5.15 | 4.80e-4 | 18.72 | 6.63e-4 | 1.99e-3 |
| AdaptDiffuser | 2.28e-4 | **4.645** | 1.76e-5 | 26.23 | 8.64e-4 | 5.49e-3 |
| Ours | **9.80e-5** | 5.01 | **8.90e-6** | **14.90** | **3.49e-4** | **8.90e-4** |

*Table 7.* Energy consumption reduction across different dynamical systems after each round of GSF. The "Before" column shows energy performance of the trained model before finetuning. For each round, "$\Delta\%$" represents the percentage reduction compared to its previous stage. Lower energy values indicate better system performance. All systems demonstrate significant initial improvements (1st round) followed by smaller incremental gains (2nd round).

| System | Before | 1st round | | 2nd round | |
|---|---|---|---|---|---|
| | | Value | $\Delta\%$ | Value | $\Delta\%$ |
| Burgers | 8.39 | 5.23 | 37.7% | 5.01 | 4.2% |
| Kuramoto | 17.48 | 15.20 | 13.0% | 14.90 | 2.0% |
| Swing | 0.27 | 0.24 | 13.4% | 0.23 | 2.4% |
| IP | 9.00e-3 | 9.50e-4 | 89.4% | 8.90e-4 | 6.3% |
| Average $\Delta\%$ | – | – | 38.4% | – | 3.7% |

optimality, while subsequent rounds primarily refine these improvements with diminishing returns. The slowing rate of improvement suggests the model is approaching a convergence point in its exploration of the optimal control space. Overall, the above result demonstrates the effectiveness of GSF in enabling exploration beyond initial training data and facilitating convergence toward optimal control strategies.

# G. Visualization

We present some visualization results of our method and best-performing baselines under three systems. The goal is to make the end state (T=10 for Burgers and T=15 for Kuramoto) close to the target state. As can be seen, SEDC's final state always coincides with the target state. In contrast, the baselines showed inferior results, as some mismatch with the target state can be observed.

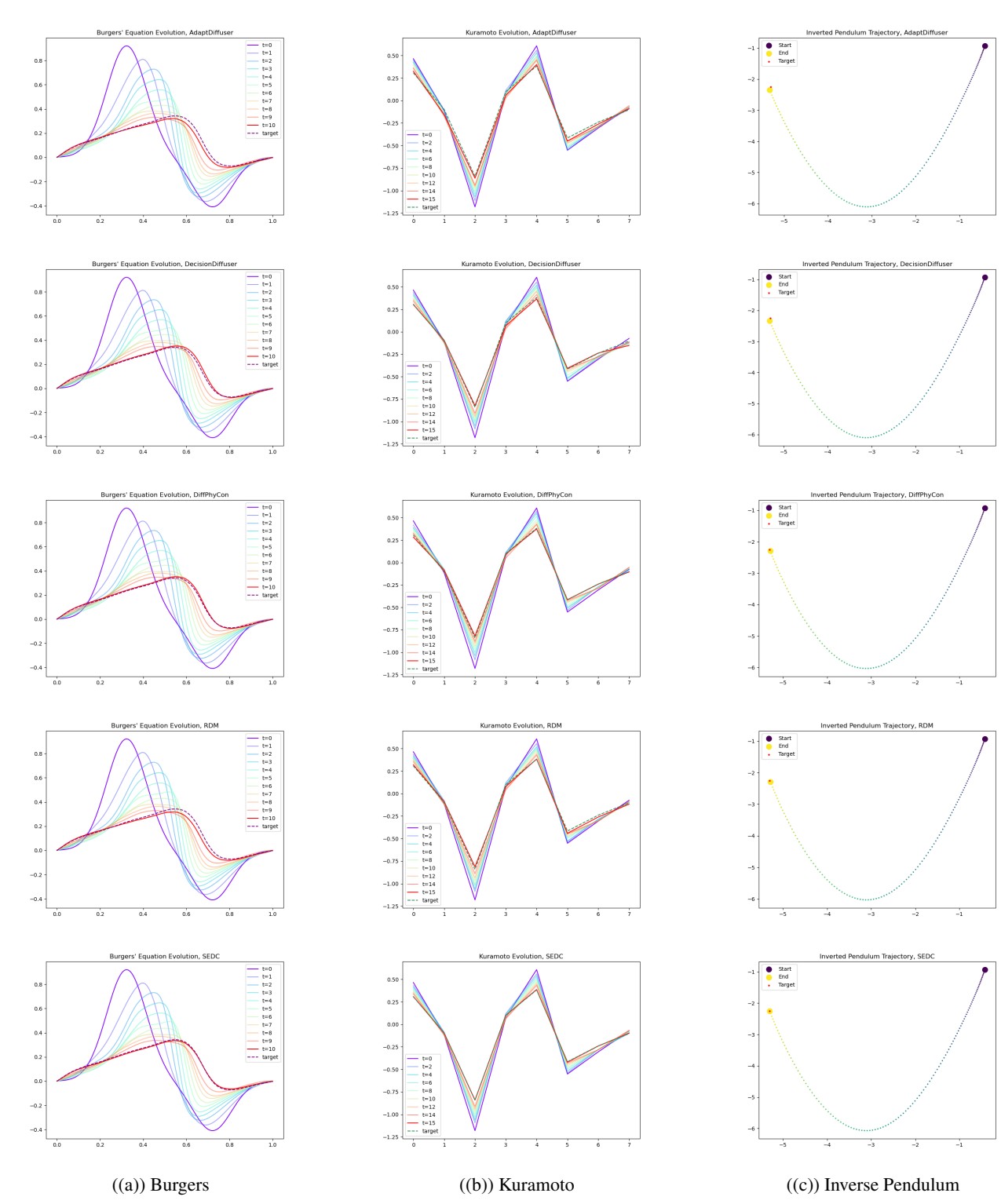

((a)) Burgers          ((b)) Kuramoto          ((c)) Inverse Pendulum

*Figure 5.* Comparison of different methods on Burgers, Kuramoto and Inverse Pendulum systems

