# OpenReview forum: "Sample-efficient diffusion-based control of complex nonlinear systems"
_ICML.cc/2025/Conference — Submitted to ICML 2025_

### Official Review · Reviewer_HvXN · 2025-03-09

**Overall Recommendation:** 2

**Summary:**

The paper presents SEDC, a new approach to improving how we control complex systems using limited data. Traditional methods struggle with high-dimensional spaces, nonlinear behaviors, and the challenge of learning from imperfect training data. SEDC tackles these problems with three key ideas: Decoupled State Diffusion, which separates state prediction from action generation to make learning more efficient; Dual-Mode Decomposition, which splits system dynamics into linear and nonlinear components for better modeling; and Guided Self-finetuning, which refines control strategies over time by generating improved training data. Experiments show that SEDC improves control accuracy over existing methods while needing less training data.

**Claims And Evidence:**

The claims in the submission are generally supported by experimental results, but some aspects require further clarification. The claim that SEDC improves control accuracy by 39.5%-49.4% while using only 10% of the training samples is backed by quantitative comparisons across three nonlinear systems, showing improved performance over baselines. The effectiveness of Guided Self-finetuning could benefit from further theoretical justification or broader validation.

The submission includes comparisons with PID and data-driven methods but should also evaluate optimal control approaches like Model Predictive Control (MPC) and Linear Quadratic Regulators (LQR). These methods are widely used for nonlinear systems and offer strong theoretical guarantees. Comparing SEDC with MPC, which optimizes control inputs over a finite horizon, and LQR, which minimizes a quadratic cost function, would provide a more comprehensive benchmark. Additionally, methods like Hamilton-Jacobi-Bellman control or Pontryagin’s Minimum Principle could offer further insights. Including these would strengthen the claims and clarify SEDC’s position among control strategies.

**Essential References Not Discussed:**

The paper builds on prior work in diffusion-based control, reinforcement learning, and nonlinear system optimization but lacks connections to optimal control methods.

**Experimental Designs Or Analyses:**

The experimental design is generally well-structured, but there are some concerns about its validity and completeness. The paper evaluates SEDC on three nonlinear systems (Burgers, Kuramoto, and Inverted Pendulum), which provide a reasonable benchmark, but it lacks real-world datasets or more diverse nonlinear control tasks to test generalizability. The comparisons with baselines, including PID, reinforcement learning, and diffusion-based methods, are useful, but the absence of optimal control methods leaves gaps in the evaluation. While the paper reports improvements in control accuracy and sample efficiency, it does not thoroughly analyze potential trade-offs, such as computational cost, training stability, or sensitivity to hyperparameters. Additionally, the fine-tuning process relies on self-generated data, but the impact of compounding errors or overfitting to model-generated trajectories is not examined. A more robust evaluation with additional baselines, real-world validation, and deeper analysis of computational efficiency would strengthen the experimental soundness.

**Methods And Evaluation Criteria:**

The methods proposed in the paper lack clear motivation and detailed explanations for key design choices, making it difficult to fully understand their necessity and effectiveness. For instance, the rationale for isolating linear and nonlinear system components is unclear—while nonlinear decomposition is a common approach in control theory, the paper does not sufficiently explain why this improves the performance of diffusion-based methods or how it compares to alternative strategies. Similarly, the use of gradient guidance to refine control trajectories is not well justified; while gradients can theoretically provide optimization signals, it is unclear how they interact with the diffusion process or whether they introduce stability issues. The fine-tuning process is also confusing, particularly how the model reuses previously generated control sequences. The paper suggests that the generated trajectories are fed back into training, but it does not clarify whether this introduces compounding errors or biases. A clearer breakdown of the fine-tuning mechanism, its impact on sample efficiency, and how it avoids overfitting to its own generated data would help in understanding its effectiveness. More intuitive explanations, ablation studies, or comparisons to alternative fine-tuning approaches would make these methods easier to evaluate.

**Other Comments Or Suggestions:**

N/A

**Other Strengths And Weaknesses:**

The definition of the symbol y in the paper is unclear and inconsistent, leading to confusion about whether it represents the observed state, system state, or observed output. In control theory, the system state refers to the internal variables that fully describe the system's dynamics, while the observed state or output is what is measurable from the system, which may be a function of the true system state. The paper seems to use y interchangeably as both the observed state and the system state, which is problematic because in many systems, the observed state does not directly correspond to the full system state. A clearer distinction between the true system state, the observed variables, and the control input is needed to avoid ambiguity. Definitions should explicitly clarify whether y is the full internal state of the system or just the observable portion and how it relates to the system’s evolution equations.

**Questions For Authors:**

N/A

**Relation To Broader Scientific Literature:**

The paper builds on prior work in diffusion-based control, reinforcement learning, and nonlinear system optimization but lacks connections to optimal control methods.

**Theoretical Claims:**

The paper lacks formal proofs for its theoretical claims, relying mainly on empirical results.

---

> ### Author Rebuttal · Authors · 2025-04-01
>
> We sincerely appreciate the reviewer's feedback. Our responses are as follows.
>
> Tip: Please visit the link(https://drive.google.com/file/d/1VWaCyEv0NPMPPqCdVfgJDPXoTiuN76MV/view?usp=sharing) for new Tables and Figures.
>
> **1. New optimal control baseline**
>
> We additionally compare SEDC with learning-based Model Predictive Control (MPC), the only data-driven method among suggested optimal control approaches. Results (link:Table I) show MPC achieves higher target losses across all tasks, likely due to error accumulation. MPC also has significantly longer inference times (e.g. >1000 vs. 0.5) that increase with control horizon. SEDC directly maps initial/target states to complete control trajectories, avoiding compounding errors and reducing computation time.
>
> **2. The rationale for isolating linear and nonlinear components in DMD**
>
> The design decomposes the clean trajectory prediction into linear and nonlinear components to overcome the limitations of single-network approaches that struggle to model both simultaneously with limited data. It is theoretically grounded in the Taylor expansion of vector-valued functions. Please refer to part 1\&2 of our response to reviewer **YCVy** for the explanation of DMD. We compare this to alternative strategies in the ablation studies. Table 1 shows that compared to single-UNet, the dual-mode architecture reduces error by 54-94\% when using only 10\% of data, and by 47-57\% with full data, confirming higher effectiveness under data scarcity. In Table 3, we also verified that as the nonlinearity of the system increases, the performance benefit gained by applying DMD becomes more pronounced.
>
> **3. The use of gradient guidance**
>
> Our gradient guidance method incorporates control cost optimization directly into the denoising process, which steers each denoising step toward trajectories that minimize an cost function $J$(e.g. control energy), using equation (3) in the paper to guide the sampling process $p_\theta(\mathbf{x}^{k-1} | \mathbf{x}^k, \mathbf{y}_0^*, \mathbf{y}_f)=\mathcal{N}(\mathbf{x}^{k-1}; \mathbf{\mu} _\theta(\mathbf{x}^k, k, \mathbf{y}_0^*, \mathbf{y}_f), \mathbf{\Sigma}^k).$
>
> The gradient term $\nabla_{\mathbf{x}^k}J(\hat{\mathbf{x}}^0(\mathbf{x}^k))$ in equation (3) computes how changes in the current noisy state affect the cost and updates the sampled mean in a gradient-descent like way. The stability is guaranteed by setting appropriate guidance strength $\lambda$ and is proven by numerous previous works like classifier-guided diffusion (Dhariwal \& Nichol, 2021) and diffusion-based planning (Janner et al., 2022). We will make the description of gradient guidance clearer in the revised manuscript.
>
> **4. Compounding errors and overfitting of GSF**
>
> GSF ensures physical consistency by re-simulating the system with generated controls (Section 4.3), ensuring all finetuning pairs $[\mathbf{u}^0_{\text{update}}, \mathbf{y}^0_{\text{update}}]$ follow system dynamics. Complementarily, our diffusion-based method considers entire trajectories holistically, allowing these two mechanisms to work together to avoid compounding errors. Moreover, generated trajectories with updated states differ from the original training data, preventing overfitting, as confirmed by the stable validation loss (link:Figures I,II) which demonstrates no upward trend.
>
> **5. Response to the datasets used**
>
> Our benchmark selection (Inverted Pendulum, Kuramoto, and Burgers) follows established control systems research practice, chosen for real-world relevance and diverse nonlinearity and complexity:
>
> - Inverted Pendulum: state 2, control 1, timestep 128
> - Kuramoto: state 8, control 8, timestep 15
> - Burgers: state 128, control 128, timestep 10
>
> We add experiments on the power grid system (please see part 5 of our response to reviewer **YCVy**) to demonstrate our generalizability on real-world scenarios.
>
>
>
> **6. The potential trade offs**
>
> Computation costs in the form of training/inference times are reported in appendix C.2, following AdaptDiffuser; other baselines do not report computational costs. The loss curve (link:Figure I) demonstrates our method's training stability. Sensitivity tests (link:Figure III) show performance is highly affected by low diffusion steps but yields only 5-10\% improvements at higher steps, following typical diminishing returns in diffusion models. These demonstrations will be included in the final version.
>
> **7. The definition of the symbol $\mathbf{y}$**
>
> In our paper, we assume full state observability throughout, with y representing the complete observable system state vector. This is reasonable for our evaluated systems. We will revise for consistent terminology and explicitly state this assumption.

---

### Official Review · Reviewer_vifZ · 2025-03-13

**Overall Recommendation:** 5

**Summary:**

This paper introduces SEDC (Sample-Efficient Diffusion-based Control), a novel diffusion-based framework designed for controlling complex nonlinear systems while addressing key challenges in sample efficiency, high-dimensional state-action spaces, and non-optimal training data. The proposed approach incorporates three major innovations: Decoupled State Diffusion (DSD) to improve efficiency in high-dimensional systems, Dual-Mode Decomposition (DMD) to enhance learning of nonlinear system dynamics, and Guided Self-Finetuning (GSF) to bridge the gap between suboptimal training data and near-optimal control policies. The model achieves remarkable performance improvements, demonstrating 39.5%-49.4% better control accuracy than baselines while using only 10% of the training data. Experiments across three nonlinear systems—Burgers dynamics, Kuramoto dynamics, and the Inverted Pendulum—validate the effectiveness of the proposed framework.

**Claims And Evidence:**

The paper makes five key claims: (1) the ability to handle high-dimensional state-action spaces (2) effective learning of nonlinear system dynamic, (3) overcoming the lack of optimal control training data by generating synthetic data, (4) achieving significant improvements in control accuracy, and reducing training data requirements and energy consumption. These claims are strongly supported by extensive experiments. They have particularly demonstrated the role of DMD structure in handling high dimensionality, DSD in effective learning of nonlinear systems, and GSF in energy efficiency through ablation studies.

**Essential References Not Discussed:**

N/A

**Experimental Designs Or Analyses:**

The experimental design is robust and methodologically detailed. The study provides extensive information about the systems and datasets used, making it highly reproducible. The experiments systematically compare SEDC to existing methods across multiple nonlinear systems, ensuring comprehensive evaluation. Various key metrics are investigated, including control accuracy, sample efficiency, and energy consumption, providing a well-rounded assessment of the model’s performance. The study is further enhanced by clear and informative visualizations that effectively highlight the advantages of SEDC over benchmarks. The ablation studies are particularly impressive, significantly strengthening the paper’s impact by explicitly demonstrating the benefits of each architectural component, thereby validating the necessity of the proposed innovations.

**Methods And Evaluation Criteria:**

The methods are well-structured, intuitive, and supported by strong theoretical grounding. The evaluation benchmarks SEDC against classical, reinforcement learning, and diffusion-based baselines, including PID, BC, BPPO, DecisionDiffuser, AdaptDiffuser, RDM, and DiffPhyCon. Performance is assessed using standard control metrics, such as Target Loss (MSE between predicted and actual target states) and Energy Consumption (integral of control effort over time). The selection of evaluation criteria is appropriate and effectively demonstrates the advantages of SEDC.

**Other Comments Or Suggestions:**

N/A

**Other Strengths And Weaknesses:**

This paper is exceptionally well-written and well-organized, with a clearly explained and well-visualized model architecture. As detailed in the Experimental Designs or Analyses section, the experimental studies are extensive and highly convincing. Additionally, the work has a broad range of applications, making it both impactful and valuable to the research community. Overall, recommend clear acceptance.

**Questions For Authors:**

Are there any plans to extend this work to stochastic control settings?

**Relation To Broader Scientific Literature:**

This paper is well-positioned within the broader literature on data-driven control, advancing diffusion-based models by building on works like DecisionDiffuser and DiffPhyCon while overcoming their limitations in sample efficiency and nonlinearity handling. Given the widespread applications of nonlinear system control and the strong performance demonstrated, this work has significant potential for impact.

**Theoretical Claims:**

N/A

---

> ### Author Rebuttal · Authors · 2025-04-01
>
> We sincerely thank Reviewer vifZ for thorough review and strong recommendation. We greatly appreciate your positive assessment of our work, particularly your recognition of our experimental design, ablation studies, and the potential impact of our work.
>
> **Response to extending SEDC to stochastic control settings:**
>
> Thank you for your question on extending our work to stochastic control settings. This represents an interesting direction for future research. Our present work focuses on deterministic non-linear systems where SEDC demonstrates significant advantages in sample efficiency and control accuracy. Although we believe the diffusion-based nature of our approach provides a conceptual foundation that could potentially be adapted to stochastic settings, this would require substantial theoretical modifications to our framework components (DSD, DMD, and GSF).
> Extending to stochastic control would involve addressing additional complexities in modeling state transition probabilities and optimizing over distributions rather than deterministic trajectories. This remains an open research question we are interested in exploring. We will add a brief discussion of these potential extensions in our limitations section to acknowledge the current deterministic focus of our work.
>
> We thank the reviewer again for their valuable feedback and encouraging assessment of our contribution.

---

### Official Review · Reviewer_YCVy · 2025-03-13

**Overall Recommendation:** 3

**Summary:**

1. The paper proposes a diffusion-based controller for high-dimensional nonlinear systems.

2. A diffusion model is used to generate a sequence of states y, and an additional autoregressive MLP is used for learning the control inputs through inverse dynamics.

3. Gradient-guidance during the reverse process and inpainting are used to satisfy an optimal control objective.

4. A dual UNet-based denoising network is proposed to decouple linear and non-linear terms of the system dynamics.

5. Experiments indicate the proposed method achieves lower target loss than baseline methods on different systems.

**Claims And Evidence:**

1. Empirical Improvements: The reported improvements in target loss and energy cost are substantial. However, many of the core innovations—gradient guidance and in-painting for goal conditioning—are direct adaptations of known methods from diffusion models and image generation (e.g., DecisionDiffuser and RePaint).

2. Concerns on Novelty: The novelty appears incremental. While the integration of these modules yields performance gains, the lack of fundamentally new theoretical insights or novel decomposition guarantees is a significant drawback.

**Essential References Not Discussed:**

n/a

**Experimental Designs Or Analyses:**

Please see points 3 and 4 in Methods and Evaluation Criteria.

**Methods And Evaluation Criteria:**

1. DMD and mode decomposition: The denoising network uses a DMD inspired architecture to predict clean states from conditions y_c and noisy trajectory x^k. The notion of decomposing a noise-corrupted state trajectory into modes is lacking clarity. In theory, for the simplified objective (eq 14) in [1], the denoising network learns to predict the noise \epsilon. The authors directly predict the clean state (\hat{x^0}) as in most implementations, however this raises question on using DMD in this case. What does it mean to decompose a noise-corrupted state trajectory into modes?

[1]: Ho, Jonathan, Ajay Jain, and Pieter Abbeel. "Denoising diffusion probabilistic models." Advances in neural information processing systems 33 (2020): 6840-6851.

2. Sample Efficiency Experiments: A critical point for sample efficiency is whether the baselines (such as DecisionDiffuser, AdaptDiffuser, etc.) were allowed to perform any form of iterative fine-tuning during inference as SEDC does with GSF. If the baselines were not fine-tuned or augmented similarly, then the comparisons might be biased. The paper should clarify if all methods underwent comparable adaptation procedures; if not, the experiment is inherently flawed. If the baselines were finetuned, the authors should clearly mention the procedure in detail for clarity.

3. Target loss calculations: The authors seem to report performance on a single held-out test set. For control applications, especially when working with limited data, it is standard practice to use cross-validation or multiple train-test splits to ensure that the reported gains are not due to a particular split. I suggest the authors to report the mean and standard deviation of the performance metrics across multiple test seeds.

4. Figure 2 appears to be plotted with a specific value of the guidance strength \lambda which controls the satisfiability of the energy constraint. This does not give the full picture of the performance of the proposed method. To compare the ability of different methods to optimise an objective with an additional constraint, it is standard practice to compare the Pareto frontiers of the methods. This would bring out the ability of the methods to optimise the objective (target loss) while having the energy constraint.

5. The paper attempts to address high-dimensional problems, however, considers the inverted pendulum and kuramoto dynamics as benchmarks. Although the inverted pendulum is considered as a classic benchmark in control theory and robotics due to its nonlinearity and underactuation, it remains a comparatively low-dimensional problem. The kuramoto model is also often considered a canonical model for studying synchronisation phenomenon and often does not capture the full complexity encountered in real-world nonlinear control problems. To demonstrate the success of the proposed method, I strongly recommend the authors to consider some well-known high-dimensional control tasks, for e.g.: a. Ant (105 states, 8 controls), Humanoid (348 states, 17 controls); from MuJoCo. b. Adroid or Shadow Hand tasks from gymnasium-robotics.

**Other Comments Or Suggestions:**

1. Writing can be improved in several places, for eg:

“We denote that x^k represents … ” can simply become “We denote x^k as the sequential data…”

2. This paper has some grammatical errors and would benefit from thorough proof-reading.

**Other Strengths And Weaknesses:**

Please see Claims and Evidence

**Questions For Authors:**

Please see Methods and Evaluation Criteria

**Relation To Broader Scientific Literature:**

The paper attempts to propose a data-driven method for model-based control of high-dimensional systems.

**Theoretical Claims:**

1. There are no theoretical contributions in this manuscript.

2. Please see point 1 in Methods and Evaluation Criteria.

---

> ### Author Rebuttal · Authors · 2025-04-01
>
> We sincerely appreciate the reviewer's feedback. Our responses are as follows.
>
> Tip: Please visit https://drive.google.com/file/d/1JmK5ZuMIg0CJCf1L2fQqobK6gtueOgts/view?usp=sharing for new tables and figures.
>
> **1.The core innovations**
>
> The gradient guidance and in-painting for goal conditioning aren't our core innovations. The innovation is a new data-driven complex system control framework that significantly improves diffusion model sample-efficiency in high-dimensional nonlinear system control. To tackle the challenge of nonlinearity of complex systems, we innovatively propose DMD that decomposes trajectory prediction into linear and nonlinear components for better capturing nonlinearity under data scarcity. The design is novel and theoretically based on the Taylor expansion. To handle high-dimensional complex physical systems, we are the first to incorporate inverse dynamics into diffusion-based control, maintaining physical consistency in trajectory generation with limited training data. Moreover, to address the unique non-optimal data problem in complex physical systems, we novelly propose GSF that enables exploration beyond the initial training data distribution.
>
> **2.Explanation of decomposition into modes**
>
> We are not decomposing a noise-corrupted state trajectory into modes. Instead, the DMD actually decomposes the prediction of the clean sampled trajectory into linear and nonlinear modes, overcoming the limitations of single-network approaches that struggle to model both simultaneously.
> The theoretical foundation is as follows:
> $\mathbf{y}_c$ is the conditional input, a learnable combination of initial and target states as in the paper. Our denoiser is designed to output the clean state trajectory $\hat{\mathbf{x}}^0$, expressed as a vector function $\mathbf{f}(\mathbf{y}_c)$. It admits a vector Taylor expansion at $\mathbf{y}_c=\mathbf{0}$:
>
> $$
> \hat{\mathbf{x}}^0=\mathbf{f}(\mathbf{y}_c) = \underbrace{\mathbf{C}_1 \mathbf{y}_c} _{\mathbf{O}_1:\text{1st-order}} + \underbrace{\mathbf{y}_c^T \mathbf{C}_2 \mathbf{y}_c} _{\mathbf{O}_2:\text{2nd-order}} + \mathcal{O}(||\mathbf{y}_c||^3)
> $$
>
> For linear systems, only the first-order term remains. For nonlinear systems, by neglecting higher-order terms for simplicity, we can decompose the prediction into linear and nonlinear quadratic modes. In our dual-Unet architecture, the first UNet learns to produce linear coefficient ($\mathbf{C}_1$) from noisy trajectories $\hat{\mathbf{x}}^k$, while the second extracts nonlinear modes ($\mathbf{C}_2$) that capture higher-order interactions.
> It is validated that as the nonlinearity of the system increases, the dual-Unet achieves more benefits compared to the single-Unet(Table 3).
> We will refine the explanation of DMD in the final manuscript.
>
> **3.sample efficiency experiment**
>
> Our experimental design is fair-our baselines include AdaptDiffuser which uses fine-tuning yet performs worse(e.g. Figure 3 shows our method achieves lower target loss with just 10\% of training data). Unlike AdaptDiffuser, we collect generated trajectories for fine-tuning without reward/discriminator filtering, exposing the model to more diverse samples and better balancing exploration/exploitation. Following your advice, we test applying GSF on the most competitive baseline DiffPhyCon. Result(link:Table III) confirms GSF improves baseline performance, validating the generalizability of our GSF framework.
>
> **4.Experiment demonstration**
>
> Similar to DecisionDiffuser and DiffPhyCon, we report mean/standard deviation metrics(link:Table I) and pareto frontiers(link:Figure I). Results confirm our method maintains competitive performance across all datasets. We will refine this demonstration in the final manuscript.
>
> **5.Limitation of the problems chosen**
>
> Our benchmark selection follows established research practice(DiffPhyCon,[1][2]), chosen for diverse nonlinear characteristics and real-world relevance. Although the Inverted Pendulum is low-dimensional, our evaluations in Burgers system(128 states/controls) have sufficiently represented our method's performance in high dimensional dynamics.
>
> Since Kuramoto may oversimplify real-world complexity, we conduct additional experiments on **swing dynamics** [1,3], which models real-world power grid behavior with higher fidelity and complexity(see link:Figure II). Results (link:Table II) show our method achieves lowest target loss, outperforming DecisionDiffuser by 60%, confirming that our approach's benefits extend to practical complex scenarios.
>
> [1] Data-driven control of complex networks. Nature Communications
>
> [2] Closed-loop Diffusion Control of Complex Physical Systems. ICLR25
>
> [3] How dead ends undermine power grid stability. Nature communications
>
> **6.Problems in the supplementary materials**
>
> We fix the problem of version compatibility error of the codes in the code link. We also report the frequency at which the benchmark systems can be operated using the methods in link:Table IV.

---

> > ### Comment · Reviewer_YCVy · 2025-04-04
> >
> > Thanks for the explanations, in particular point 2. I will raise my score.

---

> > > ### Author Response · Authors · 2025-04-09
> > >
> > > Thank you for your kind feedback and suggestions. We are glad that our rebuttal has addressed your concerns and deeply appreciate the raised score. We will incorporate them into the final paper following your advice.
> > >
> > > Thank you again for your time and consideration.

---

### Official Review · Reviewer_qYwS · 2025-03-19

**Overall Recommendation:** 4

**Summary:**

The paper presents SEDC, a novel diffusion-based control framework designed to achieve sample-efficient and robust control of complex nonlinear systems. SEDC is developed to overcome challenges associated with high-dimensional state–action spaces, strongly nonlinear dynamics, and the scarcity of optimal training data.

Experimental results across several benchmark systems—including Burgers, Kuramoto, and inverted pendulum dynamics—demonstrate that SEDC achieves 39.5%–49.4% improvement in control accuracy over state-of-the-art baselines while requiring only 10% of the training samples. Additional ablation studies confirm the effectiveness of each key component.

**Claims And Evidence:**

The claims made by the paper, from my perspective, are well-supported by its thorough numerical evaluations.

**Essential References Not Discussed:**

I am not very familiar with the literature related to this paper so I am not sure if the literature review of the paper needs any improvement or not. But it seems to me that the authors have successfully compared their methods to many existing methods in the literature.

**Experimental Designs Or Analyses:**

I think the experiments done in the paper are very thorough and the abalations are done nicely.

**Methods And Evaluation Criteria:**

The proposed method makes sense for the solving deterministic control problems and it would be interesting to see if this framework can be further extended to solve Schrondinger bridge problems or stochastic optimal control.

**Other Comments Or Suggestions:**

N/A

**Other Strengths And Weaknesses:**

N/A

**Questions For Authors:**

I don't have any questions for authors.

**Relation To Broader Scientific Literature:**

N/A

**Theoretical Claims:**

The paper does not make any significiant theoretical claims.

---

> ### Author Rebuttal · Authors · 2025-04-01
>
> We sincerely thank Reviewer qYwS for positive assessment of our work and thorough review.
>
> **Response to extending SEDC to stochastic control problems:**
>
> We appreciate the reviewer's insightful suggestion regarding potential extensions to Schrödinger bridge problems and stochastic optimal control. This represents an interesting direction we had not fully explored yet.
>
> Our framework is primarily designed for deterministic systems, and extending it to stochastic settings would require significant theoretical and architectural modifications. We believe our diffusion-based approach might provide a foundation for addressing stochastic control problems. And for further extension to stochastic scenarios,  the inverse dynamics and non-linear decomposition components have to be adapted to accommodate stochastic processes, which would require careful investigation.
> We will add a limitation section in the final paper that acknowledges the current deterministic focus of our work and discusses potential future extensions to stochastic settings as an open research question.
>
> We thank the reviewer for this valuable suggestion that opens up interesting avenues for future work.

---

### Decision · Program_Chairs · 2025-05-01

**Decision:**

Reject

**Comment:**

The paper introduces a diffusion-based approach for learning efficient control policies in complex nonlinear systems. The proposed method, SEDC, aims to address key challenges such as high-dimensional state–action spaces, strong nonlinearity in system dynamics, and limited access to optimal training data.

SEDC incorporates several components: (1) gradient guidance to steer the diffusion process toward promising trajectories (a technique previously introduced by Janner et al., 2022); (2) a direct inverse model to predict actions from states; (3) a dual UNet-based denoising network, proposed to decouple linear and nonlinear components of the dynamics (though this aspect remains difficult to interpret, despite the authors’ explanation to Reviewer YCVy); and (4) a Guided Self-Finetuning (GSF) mechanism, where generated action sequences are used to simulate new trajectories that are then added to the training dataset. Empirically, SEDC appears to perform well across three tested environments.

One major concern is the limited technical novelty of the proposed method. The use of gradient guidance has already been established in prior work, and the description of the dual UNet component lacks clarity, making it hard to discern the actual contribution. Moreover, the GSF step—while conceptually sound—relies on having access to a model of the system dynamics, which is not always feasible in practice. As confirmed in Appendix A, the model of the dynamics is an input to the algorithm. This is what created the confusion for Reviewer HvXN.

A second concern relates to the experimental validation. Two of the three tested environments are relatively low-dimensional and are standard benchmarks in the control literature, which weakens the claim that the method addresses high-dimensional settings. Additionally, the paper does not compare SEDC against classical control approaches such as Model Predictive Control (MPC) or data-driven MPC, despite their strong performance in similar settings (e.g., inverted pendulum). Although the authors provided additional experiments with MPC during the discussion phase, the comparison remains incomplete and should be significantly expanded. Note the code has not been made available, the link provided does not work.